Coral micro-fragmentation assays for optimizing active reef restoration efforts

http://orcid.org/0000-0003-1279-8673 Knapp Ingrid S.S. 1 ingrid.knapp16@gmail.com
http://orcid.org/0000-0002-5837-3614 Forsman Zac H. 1 2
Greene Austin 1
Johnston Erika C. 1 3
Bardin Claire E. 1
Chan Norton 4
Wolke Chelsea 4
Gulko David 4
http://orcid.org/0000-0001-6339-4340 Toonen Robert J. 1 rjtoonen@gmail.com
1 Hawai'i Institute of Marine Biology, School of Ocean & Earth Sciences & Technology, University of Hawai'i at Mānoa , Moku o Lóe, Kānéohe, Hawai'i , USA
2 Environmental Science and Monitoring, The Red Sea Development Company , Riyadh , Saudi Arabia
3 Department of Biological Science, Florida State University , Tallahassee, Florida , United States
4 Hawai'i Coral Restoration Nursery, Hawai'i Division of Aquatic Resources , Honolulu, Hawai'i , USA
Bieler Rüdiger
Electronic publication date: 2022 Jul 18
Publication date: 2022
Volume: 10
Electronic Location ID: e13653
Received 2022 Jan 27; Accepted 2022 Jun 8
Copyright: © 2022 Knapp et al.
Copyright year: 2022
Copyright holder: Knapp et al.
License: This is an open access article distributed under the terms of the Creative Commons Attribution License, which permits unrestricted use, distribution, reproduction and adaptation in any medium and for any purpose provided that it is properly attributed. For attribution, the original author(s), title, publication source (PeerJ) and either DOI or URL of the article must be cited.
License URL: https://creativecommons.org/licenses/by/4.0/

Keywords: Hawai'i, Montipora capitata, Porites lobata, Coral nursery, Outplanting, Fragment size, Residence time, Structure from motion, Net growth, Survivorship

Funding: National Oceanographic and Atmospheric Administration (NOAA) Coral Reef Conservation Program (CRCP) NA18NOS4820110 NOAA National Marine Fisheries Service (NMFS) Pacific Islands Regional Office (PIRO) NOAA-NFA-NFAPO-2018-2005418 This work was supported by the National Oceanographic and Atmospheric Administration (NOAA) Coral Reef Conservation Program (CRCP) grant (No. NA18NOS4820110) and the NOAA National Marine Fisheries Service (NMFS) Pacific Islands Regional Office (PIRO) grant (No. NOAA-NFA-NFAPO-2018-2005418).

==============================
The global decline of coral reefs has driven considerable interest in active coral restoration. Despite their importance and dominance on mature reefs, relatively few coral restoration projects use slower growth forms like massive and encrusting coral species. Micro-fragmentation can increase coral cover by orders of magnitude faster than natural growth, which now allows cultivation of slow growing massive forms and shows promise and flexibility for active reef restoration. However, the major causes of variation in growth and survival of outplanted colonies remain poorly understood. Here, we report simple outplanting assays to aid in active reef restoration of slower growing species and increase the likelihood of restoration success. We used two different micro-fragmentation assays. Pyramid assays were used to examine variation associated with fragment size (ranging from ≈1–9 cm2), nursery residence time (for both in-situ and ex-situ nurseries), and 2D vs. 3D measurements of growth. Block assays were used to examine spatial variation among individual performance at outplanting sites in the field. We found 2D and 3D measurements correlated well, so measured survivorship and growth using top-down planar images for two of the main Hawaiian reef building corals, the plating Montipora capitata and the massive Porites compressa. Pyramid assays housed and outplanted from the in-situ nursery showed no effect of residence time or size on overall survivorship or growth for either species. Results from the ex-situ nursery, however, varied by species, with P. compressa again showing no effect of nursery residence time or size on survivorship or growth. In contrast, nursery culture resulted in improved survivorship of small M. capitata fragments, but net growth showed a weak positive effect of nursery time for medium fragments. Small fragments still suffered higher mortality than either medium or large fragments. Due to their lower mortality, medium fragments (≈3 cm2) appear to be the best compromise between growth and survivorship for outplanting. Likewise, given weak positive gains relative to the investment, our results suggest that it could be more cost-effective to simply outplant medium fragments as soon as possible, without intermediate culture in a nursery. Furthermore, the block assay revealed significant differences in survivorship and growth among sites for individuals of both species, emphasizing the importance of considering spatial variation in coral survival and growth following outplanting. These results highlight the value of using short-term micro-fragmentation assays prior to outplanting to assess size, and location specific performance, optimizing the efficiency of active reef restoration activities and maximizing the probability of success for active coral restoration projects.

Introduction

As coral reef ecosystems continue to decline worldwide, many have called for active intervention and innovative management tools to address conservation challenges and reverse the decline of coral reef habitats (Anthony et al., 2017; Kleypas et al., 2021; Rinkevich, 2005; van Oppen et al., 2015, 2017; Vardi et al., 2021; Vaughan, 2021). Corals form the structure and foundation of coral reefs, fulfilling an ecosystem engineering role analogous to trees in terrestrial ecosystems (Quigley, Hein & Suggett, 2022). The ethics and scalability of active interventions to reverse coral reef decline remain a subject of debate (Anthony et al., 2020; Caruso, Hughes & Drury, 2021; Doropoulos et al., 2019; Filbee-Dexter & Smajdor, 2019; Williams et al., 2018), but are common management strategies among terrestrial ecosystems. For example, one of the most widely used conservation and management tools for forests is to incorporate a nursery phase where vulnerable seeds, saplings, or propagules are sheltered and provided conditions to greatly increase the probability of survivorship, a strategy that has dramatically transformed forest ecosystems (Fox, Jokela & Allen, 2004; Khurana & Singh, 2001). Over the last two decades, coral nurseries have transitioned from small scale pilot projects, to large scale operations dedicated to production for the marine hobby industry (Delbeek, 2001; Tlusty, 2002), the conservation of rare or endangered coral species (Griffin et al., 2012; Herlan & Lirman, 2009), or active coral reef restoration (Boström-Einarsson et al., 2020; Epstein, Bak & Rinkevich, 2003; Nedimyer, Gaines & Roach, 2011; Rinkevich, 2008).

The potential benefits of reef restoration activities vary from site to site, because natural recruitment and recovery rates are highly variable, both temporally and spatially (Connell, Hughes & Wallace, 1997; Kojis & Quinn, 2001). Some reefs surrounded by high coral cover might naturally recover from disturbance within a period of decades (Adjeroud et al., 2009; Connell, Hughes & Wallace, 1997; Jury & Toonen, 2019), whereas other reef systems may take an order of magnitude longer if they ever recover at all (Hughes & Tanner, 2000; Salinas-de-León et al., 2013; Smith, 1992). Recruitment failure and high rates of post-settlement mortality of corals can result in a downward spiral of ecosystem collapse and transition to a macroalgal dominated alternative state (Dudgeon et al., 2010; Briggs et al., 2018; Hughes & Tanner, 2000). Once ecological systems transition to an alternative state, such as macroalgal dominance on coral reefs, it often requires much higher densities of herbivores to transition back than it did to maintain the previous state (Fung, Seymour & Johnson, 2011; Mumby, Steneck & Hastings, 2013; Schmitt et al., 2019). Thus, reverse transitions, from algal to coral-dominated ecosystems are rarely observed, but increased fish and coral recruitment have been documented to occur with some large scale reef restoration efforts in both the Caribbean (Huntington et al., 2017; Opel et al., 2017; Schopmeyer & Lirman, 2015) and the Indo-Pacific (Cabaitan, Yap & Gomez, 2015; Lamont et al., 2022; Yap, 2009).

Most reef restoration efforts seek to augment three-dimensional structure and live coral cover (Boström-Einarsson et al., 2020). To increase the productivity required to scale-up restoration, the success of such efforts is dependent on finding the optimal colony size and nursery residence time for outplants that maximizes effectiveness of the restoration (dela Cruz et al., 2015). Outplanting of larger coral fragments through rearing juveniles or small fragments in nurseries often translates to increased probability of survival for coral colonies (Page, Muller & Vaughan, 2018; Raymundo & Maypa, 2004; van Woesik et al., 2021). Thus, most coral nurseries seek to provide safe environments in which corals are maintained under ideal conditions prior to outplanting until their risk of mortality is reduced (by reaching a size refuge). However, it takes time to grow large fragments even under nursery conditions, and the larger the starting size, the fewer total fragments can be taken from a parent colony (Forsman, Page & Vaughan, 2021). Prolonged nursery culture not only increases labor and maintenance costs but also requires substantially more space to maintain equivalent output, which impacts scaling during restoration efforts.

Nursery costs depend not only on duration of culture, but also the type of nursery: in-situ (in the water) and ex-situ (in tanks on land) culture each have a suite of costs and benefits to consider (Vaughan, 2021). In-situ nurseries have minimal maintenance and equipment costs, but environmental conditions are more difficult to control (e.g., light, temperature, sedimentation, competition, predation, disease), whereas ex-situ nurseries maintain perfect conditions at a premium in terms of labor, setup and operational (utilities, water quality, supplies, and infrastructure maintenance) costs. One key to improving efficiency and reducing costs for both types of nurseries is to reduce the amount of time that fragments need to be maintained prior to outplanting. Thus, identifying the ideal size for outplanting success is of high value and essential to optimizing efforts to scale up restoration. However, the ideal size for trading off survivorship and costs is likely to vary both temporally and spatially, as well as among species, in the same way that individual growth rates vary in the same corals through time (Edmunds & Putnam, 2020). For example, previous studies have found relationships between size and mortality vary with nursery conditions (Forsman, Rinkevich & Hunter, 2006), habitat (Bruno, 1998), bleaching events (Depczynski et al., 2013), and competitive interactions (Ferrari, Gonzalez-Rivero & Mumby, 2012). Restoration efforts of slower growing species must optimize tradeoffs between a strategy to outplant larger coral colonies with higher survival but greater investment per individual against one of outplanting many smaller colonies with minimal investment in each (Forsman, Page & Vaughan, 2021). For example, artificial substrates seeded with new coral recruits showed a 5–18-fold reduction in out-planting costs by dramatically reducing diver labor, which is the costliest aspect of reef restoration work (Chamberland et al., 2017). Survivorship is often low among recruits and highly stochastic in coral reef ecosystems (Edmunds, Bruno & Carlon, 2004; Irizarry-Soto & Weil, 2009; Miller, 2014), so it is not surprising that only 9.6% of newly settled corals survived their first year, but this essentially offsets the initial cost savings. Because corals show variable sensitivity, both within and among species, to environmental parameters such as sedimentation, pollution, temperature, irradiance, salinity, and pH (Bahr, Jokiel & Toonen, 2015; Fabricius, 2005; Kleypas et al., 2021; Lough & Barnes, 2000; Williams et al., 2010) it is also important to determine whether there is variation in survival when attempting to scale-up restoration efforts. Therefore, prior information about genotype-and species-specific responses at a particular restoration site could maximize survival while minimizing cost and ensuring the most cost-effective approach to mass producing and outplanting corals for reef restoration. Coral micro-fragmentation (Forsman et al., 2015; Forsman, Page & Vaughan, 2021; Page, Muller & Vaughan, 2018; Vaughan et al., 2019) can precisely control colony size and genotype (donor colony) for outplanting, with the potential to develop highly flexible and cost-effective assays on site-specific data mortality and growth for replicated genotypes across a range of sizes. Micro-fragmentation is primarily an ex-situ nursery based method which results in rapid two-dimensional spreading of tissue at rates that can be orders of magnitude higher than growth rates under typical field conditions (Forsman et al., 2015, Forsman, Page & Vaughan, 2021; Page, Muller & Vaughan, 2018). The technique typically uses small (~1 cm2) fragments all from the same donor colony (and therefore same genotype) spaced approximately 2–3 cm apart over an artificial substrate. This method takes advantage of the tendency of these small fragments to rapidly grow from the cut edges (Soper et al., 2022) spreading thin layers of tissue which fuse upon contact, doubling or quadrupling their size within a few months (Forsman, Page & Vaughan, 2021). When such fragments are attached to dead coral heads they can quickly ‘re-skin’ an entire colony, which can result in bringing large endangered corals back to life (Page, Muller & Vaughan, 2018). Knowledge of genotype-and size-specific mortality rates for corals at a given site would allow restoration efforts to target mass production of resilient genotypes of an optimal size, to maximize cost-effectiveness and scale while improving the outcome of restoration.

Here, we evaluate the impacts of fragment size and nursery residence time at both an in-situ (the Hawai'i Institute of Marine Biology (HIMB) mid-water coral farm), and ex-situ (the Hawai'i Division of Aquatic Resources’ Hawai'i Coral Restoration Nursery’s (HCRN) land-based facility) coral nursery. We use that information to test spatial variation in outplanting performance across an environmental gradient then combine these approaches to propose a rapid assay approach to improve strategies to increase time-and cost-effectiveness of reef restoration efforts in the field.

Materials and Methods

Study species

We selected Montipora capitata (Family Acroporidae) and Porites compressa (Family Poritidae) for these assays. These are two of the dominant reef building coral species on O‘ahu (Fletcher et al., 2008; Franklin, Jokiel & Donahue, 2013), and represent two of the major life history categories of reef-building corals (Darling et al., 2012). M. capitata is a highly polymorphic encrusting species that forms plates and branches as it matures. P. compressa is a massive coral that forms large mounds with cylindrical branches that often fuse.

Micro-fragmentation

All experimental fragments for both assays were cut to size with a Gryphon XL Aquasaw and 42′ diamond tipped stainless steel blade, and fixed to the substrate using cyanoacrylate (Bulk Reef Supply extra thick gel superglue, Golden Valley, MN). To standardize treatment all the undersides of fragments were freshly cut flat to ensure greater adhesion, even if the coral was flat before cutting. We aimed to cut the fragments leaving little to no skeleton exposed for algal growth. If any coral skeleton was exposed, we covered it in superglue to deter predators, particularly in the case of P. compressa, which is often heavily predated on by an aeolid nudibranch (Phestilla sp.) (Faucci, Toonen & Hadfield, 2007). After gluing, but before moving the substrate, each fragment was checked to ensure it was firmly fixed to the assay to avoid any accidental loss of corals.

Experimental locations

In-situ nursery

The field-based nursery was located on Moku o Lo‘e (Coconut Island) at the Hawai'i Institute of Marine Biology (HIMB) in Kānéohe Bay (Figs. 1A, 1B) and was constructed to conduct research on improving the time and cost efficiency of reef restoration. The coral nursery consists of floating walkways surrounding and supporting suspended midwater platforms for coral cultivation. It was constructed in 2017 from recycled materials salvaged from decommissioned aquaculture and marine mammal pens. The primary source of the over 1,000 corals housed in the nursery is from retired pens and other ‘corals of opportunity’ relocated to the nursery from marine debris or other salvage which would otherwise be discarded. Collection and monitoring work was approved by the Division of Aquatic Resources (DAR) under Special Activities Permits (SAP) 2018-03, 2019-16, and 2020-25.

Figure 1 Map of in-situ and ex-situ nurseries and their respective outplanting sites for the pyramid assays.

Map of O‘ahu, Hawai'i indicating the two experimental locations, with the darker blue indicating the nurseries and the lighter blue the outplanting sites where (A) is the in-situ nursery at the Hawai'i Institute of Marine Biology (HIMB), (B) the in-situ outplanting reef site, (C) the ex-situ nursery tanks at Ānuenue Fisheries Research Center (AFRC) Hawai'i Coral Restoration Nursery (HCRN), and (D) the ex-situ outplanting reef site. Map data © 2022 Google, and TerraMetrics.

Ex-situ nursery

The Division of Aquatic Resources’ Hawai'i Coral Restoration Nursery’s (HCRN) ex-situ nursery is located at the Ānuenue Fisheries Research Center (AFRC) on Sand Island (Figs. 1C, 1D). It was built for the purpose of improving methods of coral culture and outplanting, to restore and conserve Hawai'i’s coral reefs, and consists of small to large indoor and outdoor tanks with varying levels of filtration and control of temperature, lighting, water chemistry, and biotic communities. A full-time staff of professional aquarists provide daily husbandry for the maintained corals. The facility primarily outplants a range of Hawaiian species ranging in size from 15 cm to over 1 m in length. The source material for this land-based nursery is predominantly coral that would otherwise have been destroyed from various state and federal projects such as harbor improvements or dredging.

Outplanting sites

Each nursery had an adjacent natural reef outplanting site. The in-situ nursery outplanting site (Fig. 1B) was located in an enclosed bay with low water flow and composed of a sandy substrate situated next to an existing mature M. capitata and P. compressa coral reef structure with roughly 70% coral cover (Caruso et al., 2021; McGilly, 2019). This site was selected so assays could be placed on the adjacent sand flat rather than affixed to the reef. The site for the ex-situ nursery outplanting (Fig. 1D) was located in Māmala Bay on the south shore of O‘ahu with hard bottom pavement. Due to high water flow conditions here (Grigg, 1998) assays were fixed to the reef with Z-SPAR A-788 splash zone epoxy (Pettit Paint, Rockaway, NJ, USA) in open patches among live coral colonies.

Pyramid assays

Pyramid assays were developed to study the effect of fragment size (small, medium or large) and nursery residence time (0, 4 and 8 months) on coral survivorship and net growth from both in-situ and ex-situ nurseries (Fig. 1). The important role of genotypic variation (Baums, 2008; Grottoli et al., 2021) was incorporated in this experiment as a random effect with three separate colonies selected per species (M. capitata, and P. compressa) and per nursery (ex-situ and in-situ). 3D photogrammetry techniques were also examined at the in-situ nursery site due to the rapid development of fragments into arborescent forms that was not captured by 2D imaging of growth.

Assay design and deployment

The pyramid assay design was a modified smaller version of the ones regularly used at the HCRN with three sides and a flat top, leaving space for a label (Fig. 2). The pyramid shape was selected because it reduces the horizontal surface upon which sediment is retained, while minimizing the 3D surface area over which corals must grow to fuse and rapidly cover the artificial substrate. Once completely covered, this design also blends well into the reef substrate which is why such coral modules were selected for outplanting by the HCRN. A polyvinyl chloride sheet (Celtec®) was cut into a form consisting of three sides and a top (Fig. 2) and joined together by drilling holes on the edge and joining them together with zip ties. The forms were nestled upside-down into tubs of sand and a fast-setting concrete (Portland Type II cement) was poured into them. Once the concrete set the zip ties joining the forms sides were cut and the concrete pyramids were removed. All pyramids were soaked in seawater for a month to cure, and then allowed to air dry in the sun for an additional month. The labels were made on a Dymo® label maker and affixed with All-Fix two-part epoxy putty prior to coral fragment attachment.

Figure 2 Pyramid and block assay designs.

The two coral micro-fragmentation assays used to assess (A) variation in growth and survival of Montipora capitata and Porites compressa fragments associated primarily with size and nursery residence time. Each face of the pyramid assay held either seven small (1 cm2), three medium (3 cm2), or one large (9 cm2) coral fragment/s from the same donor colony. In order to avoid position effects on these pyramid assays each replicate (A, B or C) rotated the location of each fragment size) and were outplanted with the top label (in gray) facing the same direction in order to expose all fragment sizes to all the potential water flow and light conditions, and (B) spatial variation in outplanting performance. Each assay consisted of nine P. compressa fragments on the upper portion and nine M. capitata fragments on the lower portion of the block. No two positions were occupied by the same donor colony genotype (numbers in the circles) across the four replicates (A, B, C, and D). There were two labels (in gray) for redundancy (one on the bottom right corner on the top face and one on the top edge), in case of loss or overgrowth the assay could still be identified.

Three unique M. capitata and three P. compressa parent colonies, roughly 30 cm in diameter, were collected at each site from within or adjacent to the nursery (Figs. 1A, 1C, and Appendix 1A, 1B) resulting in six parent colonies from a combination of natural reef and coral nursery origin at each location. The standard quarantine period was also observed for the ex-situ nursery samples, whereby any parent colonies not already in the nursery had all epifauna removed before being placed in a quarantine tank where they were required to remain clear of aquatic invasive species (AIS), parasites or visible disease for at least 1 month before experimentation (Appendix 1A, 1B). Coral predatory nudibranchs (Phestilla sp.) emerging during quarantine on the P. compressa parent colonies at the ex-situ nursery resulted in additional cleaning and delayed fragmentation and deployment by 2 months at this site.

The parent corals were micro-fragmented to yield seven small (≈1 × 1 cm or 1 cm2 each, 7 cm2 total), three medium (≈1.75 × 1.75 cm or 3 cm2 each, 9 cm2 total), and one large (≈3 × 3 cm, 9 cm2 total) fragment(s) per pyramid (Fig. 2). Each of the six unique colonies per site were fragmented into three identical replicates (A, B and C), which were then outplanted at 0, 4 and 8 months resulting in 27 assays per species (54 total) per reef outplanting site (see Appendix 2 for detailed timeline of coral assay deployment). The assays outplanted at 0 months (T0) had no nursery time and were outplanted immediately following confirmation of solid attachment of the fragments (Figs. 1B, 1D). The remaining two sets were kept in nursery conditions (Figs. 1A, 1C) until outplanting of the second set of 18 pyramids at the same reef sites in month 4 (T1), leaving the final replicate in the nursery. After a further 4 months (T2, 8 months since T0 was deployed) the third set was then also outplanted at the same locations.

The top (the location of the label) and all sides of the pyramid were photographed with a ruler, for scale, before placing them in the nursery or outplanting site. In order to avoid effects of position on the assay, the location of each fragment size was alternated around the pyramid with a clearly visible label on the top i.e., all ‘A’ replicates had the large fragment on the upper face, the medium on the left and the small on the right, relative to the label (Fig. 2). The pyramids were then outplanted with labels all facing the same direction, ensuring that each fragment size was exposed to all the potential variable light and water flow conditions.

In addition to recording survivorship (alive or dead), fragment size (cm2) of all 1,188 corals was manually measured from planar scaled digital photos (Fig. 3) using the program ImageJ (Schneider, Rasband & Eliceiri, 2012) at the point of fragmentation, outplanting, and at the end of the experiment. The final survivorship and growth measurements were collected at the in-situ nursery 4 months after the final deployment, but were delayed due to COVID-19 lockdowns until 9 months after final deployment at the ex-situ nursery (Appendix 2).

Figure 3 Examples of pyramid assay fragment growth from the in-situ and ex-situ nurseries, 3D structure from motion (SfM) segmentation and labeling, and the relationship between 2D and 3D SfM measurements.

Examples of Porites compressa (left) and Montipora capitata (right) medium fragments spreading horizontally and fusing and/or growing vertically on assays at both ex-situ (A, B), and in-situ (C, D) coral nurseries. An example of segmentation and labeling of living coral tissue on an in-situ nursery coral assay module for the estimation of surface area covering the complex geometry of coral colonies (E) and the relationship between two-dimensional (planar) area in cm2 to surface area derived from 3D Structure from Motion (SfM) models (F) where light grey is P. compressa and dark grey is M. capitata.

3D photogrammetric measurements

The coral fragments at the in-situ nursery (Figs. 1A, 1B) grew with a much higher degree of three-dimensional structure than those of the ex-situ nursery (Figs. 3A–3D). Therefore, we added 3D photogrammetry to estimate surface area of living coral tissue at the in-situ nursery in addition to estimating growth from planar scaled digital images that was performed at both locations.

A Three-Dimensional Structure from Motion (3D SfM) photogrammetric model of coral assays at the in-situ nursery was constructed using Agisoft Metashape Pro v 1.5.5 (AgiSoft Metashape Professional, 2019), from approximately 500 photos taken with a Canon Rebel EOS in an Ikelite underwater housing (Fig. 3E). Camera settings and assembly of the SfM model followed recommendations in Suka et al. (2019). Briefly, manual camera settings were selected (auto ISO exposure, f-stop = F10, shutter speed = 1/320, -⅓ exposure, broad point autofocus, repeat shutter, and large format photos). A batch script in Metashape Pro was run with the following settings (alignment = high accuracy and generic precision, 40k key point limit, 4k tie point limit, adaptive camera model fitting, Optimization = fit f and cx, cy, build dense cloud = medium quality, mild depth filtering, build mesh = arbitrary surface type, medium depth map quality, build texture = generic mapping mode, texture from all cameras, and hole filling enabled, build tiled model = source data dense cloud with medium depth map quality). The resulting SfM model was scaled with a series of six printed targets, fixed in pairs 10 cm apart. The accuracy of the three scale bars was 0.1 cm, with an overall estimated error of 0.07 cm. The scaled 3D SfM model was exported into Cloudcompare v2.11 (GPL Software) and areas with living coral tissue were segmented for inclusion with the segmentation tool. Corals on each side of the pyramid were grouped, labeled and colorized using an elevation model to highlight upward growth along the Z axis (Fig. 3E). Surface area was estimated for each size category (e.g., fused or unfused corals were grouped together for estimation of total surface area for each size category). Finally, we compared two-dimensional area (from top-down measurements) with three-dimensional surface area estimates from the SfM model by linear regression in R (RStudio, Inc. RsT, 2019) (Fig. 3F).

Statistical Analyses

Generalized linear mixed models (glmm) were used to assess the likelihood of individual fragments surviving to the end of the experiment based on their total time in nursery and their size class (see Appendices 3–5 for raw data and analyses). To avoid bias from fragments that did not survive their time in nursery, only fragments surviving to their designated outplanting time were included in these analyses. A glmm was fit for each species within each nursery using the glmer() function from the lme4 R package, with a binomial response defined in the model specification. Linear mixed models allowed the use of random effects to account for variation due to parent colony (genotype) and deployment pyramid within colony. Fixed effects included an interaction between experimental day outplanted (total time in nursery) and size group on the pyramid face being estimated. To ensure the reliability of glmm results we inspected model convergence as well as variance inflation using the vif() function from the R package car. Overall model performance was evaluated as adjusted marginal versus conditional R2 using the r.squaredGLMM() function from the MuMIn R package. An example glmm model specification would be:

Survival∼Day_Outplanted∗Size+(1|Genotype/Pyramid)

Percent net growth was calculated as the percent difference between total living coral tissue area at the beginning and end of the experiment and was assessed across all fragments on each face of a pyramid. Linear Mixed Models (lmm) were fit using the R package lme4 for both M. capitata and P. compressa datasets within each nursery, resulting in a total of four separate fitted models to estimate the effect of nursery time and size class on pyramid face net growth. Similar to models used previously to assess survivorship, these models included nested random effects for parent colony genotype and pyramid within genotype, in addition to a fixed effect interaction between experimental day outplanted and fragment size class. After model fitting a type three ANOVA was performed using the Anova() function in the R package car to assess the significance of marginal effects in each model. An example lmm model specification would be:

PercentNetGrowth∼Day_Outplanted∗Size+(1|Genotype/Pyramid)

Block assay

In addition to fragment size and nursery time, variation among colonies in response to the conditions at the outplanting site is a key factor to understand and improve restoration success, because no colony is resilient to every stress they may encounter among different environments. The pyramid assays were only capable of examining variation among three genotypes; therefore, we designed a second assay to specifically examine spatial variation in greater detail while accounting for high intraspecific variation in survival and growth among M. capitata and P. compressa fragments.

Ten outplanting sites throughout Kānéohe Bay, ranging from 0.5 to 3.2 m in depth, were selected (Fig. 4A) to encompass the range of environmental and hydrodynamic variability seen across the bay (Bahr, Jokiel & Toonen, 2015; Caruso et al., 2021; Ostrander et al., 2008). We used horizontal pre-formed concrete slabs (40 × 20 × 5 cm wall cap block) as “block assays” for this purpose. Four replicate medium (≈3 cm2) fragments were cut from each of nine unique and widely spaced parent colonies to ensure distinct genotypes (Appendix 6), and those fragments were used to create four replicate blocks (A, B, C and D) per site. To avoid position effects we used a random number generator to ensure no two locations on the assay blocks were occupied by the same genotype. All nine P. compressa fragments were co-located on the upper portion and all nine M. capitata fragments on the lower portion of each block to minimize potential for interspecific competition (Figs. 2 and 5A–5C). We also made sure that the bottom row of P. compressa fragments and top row of M. capitata fragments had at least one of each genotype (across all four replicates) to account for potential species interaction effects. Finally, two labels were attached to each assay (Fig. 2), one on the bottom right corner and one on the top edge, to minimize potential for loss or coral overgrowth making the label illegible.

Figure 4 Map of block assay outplanting sites within Kānéohe Bay, Hawai'i along with percent survivorship and net growth plots.

Block assay (A) map of Kānéohe Bay, O'ahu outplanting sites (1–10) from the in-situ nursery (white dot), (B) percent survivorship (colored) and mortality (gray) of fragments after outplanting for Montipora capitata and Porites compressa across sites (1–10), and (C) violin plot of percent net growth across sites (1–10) for M. capitata and P. compressa. Map data © 2022 Google, Maxar Technologies, and USGS.

Figure 5 Example of a block assay: over time, in the in-situ nursery and while outplanted, along with focus on the growth of one fragment from the beginning to the end of the experiment.

Block assay design with the top row consisting of the same assay with nine Porites compressa fragments on top and nine Montipora capitata fragments on the bottom (A) immediately after fragmentation (B) the day of outplanting (day 112), and (C) the day of retrieval (day 243). Coral assays in the nursery on racks after fragmentation (D), an outplanted coral assay in Kāne‘ohe Bay (E), and the same M. capitata fragment in imageJ used to calculate net growth (cm2) directly after fragmentation (F) and then retrieval (G).

Colonies were fragmented and attached to the blocks in February 2020 (Fig. 5D), after which they were held at the HIMB in-situ nursery for 4 months. Block assays were deployed, via snorkel, for 4 months beginning in June (Fig. 5E) and were retrieved in October 2020. Because there was minimal vertical growth, all assays were top-down planar photographed with a ruler at each time point. Twenty-eight (out of 720) fragments died in the nursery shortly after fragmentation, so they were replaced and the difference in fragmentation dates was factored into the statistical analyses. All replicates (A, B, C and D) were maintained on separate submerged racks within one nursery pen (Fig. 5D) to ensure no one outplanting site had all assays in one area of the nursery. Each submerged rack measured approximately 1 × 10 m and was constructed of PVC and plastic mesh with replicates arranged in alphabetic order. Survivorship and net growth (cm2) were documented for all samples after fragmentation, before deployment, and after retrieval. As with the pyramid assays, survivorship was recorded in the field as a binary response, either alive (1) or dead (0), and net growth (cm2) was measured from planer top-down scaled digital photos in ImageJ (Figs. 5F, 5G).

Statistical Analyses

We define survivorship as those fragments which survived after deployment to the reef site until they were retrieved, therefore corals that died in the nursery prior to deployment were excluded. In order to incorporate both the fixed effect of ‘site’ and the random effect of ‘genotype’ (parent colony) we analyzed the binary response of survivorship using glmm models with a binomial (logit link) error distribution from the lme4 package (Bates et al., 2015) in RStudio (See Appendices 7 and 8 for raw data and analyses). Similar to the pyramid assays, variance inflation was inspected and overall model performance was assessed as adjusted marginal vs. conditional R2.

Survival∼Site+(1|Genotype)

Percent net growth included only those fragments which survived to the end and was calculated as 100*(cm2 growth at the end/cm2 growth at the beginning), which factors in the differing initial sizes of the coral fragments that had been grown for variable periods in the nursery prior to field deployment. We used lmm models in the lme4 package to analyze percent net growth with site as a fixed and genotype as a random effect.

PercentNetGrowth∼Site+(1|Genotype)

Net growth for P. compressa was square-root transformed as required to meet assumptions of normality and homoscedasticity, but M. capitata did not require transformation as determined by Q-Q plots, histograms, and residuals over fitted plots. Similar to the pyramid assays after model fitting a type three ANOVA was performed using the Anova() function.

Results

Pyramid assays: fragment size and nursery residence time

Fragment survival likelihood among sizes and residence times

Overall, small fragments were significantly less likely to survive to the end of the experiment (42%) compared to medium or large fragments (67% and 70%, respectively) which were not significantly different from one another (Appendices 9–11). GLMM models of individual fragment survivorship indicated that survivorship of M. capitata fragments at the ex-situ nursery outplanting site was significantly improved through increased time in the nursery and this effect was greatest among the smallest fragments (Appendix 10A). By comparison, there was no significant effect of nursery time on P. compressa survivorship for the ex-situ nursery outplanting site (Fig. 6 and Appendix 10B). A large proportion of survivorship variance was attributed to the nested effect of pyramid within genotype with approximately 80.8% (40.7% marginal) for M. capitata and 68.2% (13.9% marginal) of variation for P. compressa explained (Appendix 10). In contrast to the ex-situ outplanting site, there were no significant effects of nursery time, size class, or the interaction of these effects for fragment survivorship at the in-situ outplanting site for either coral species (Appendix 11). A positive, but marginally insignificant, effect was observed for the interaction between medium size and nursery time for fragments of Porites compressa at the in-situ nursery outplanting site. For the in-situ outplanting site, more of the variation in survivorship was attributed to fragment genotype than to pyramid within genotype for M. capitata, whereas survivorship of P. compressa varied more with pyramid within genotype. (Appendix 11).

Figure 6 Pyramid assay percent net growth and survivorship of small, medium, and large fragments relative to outplanting time for Montipora capitata and Porites compressa housed at the in-situ and ex-situ nurseries.

Pyramid assay percent net growth (boxplots) and percent survivorship (lines) of Montipora capitata and Porites compressa by fragment size (small (≈1 cm2), medium (≈3 cm2) and large (≈9 cm2)) at the end of the experiment based on the time (T) they were outplanted (T0, 1, and 2), which were either 0, 111, and 254 days for the ex-situ nursery (in blue), or 0, 116 and 250 days for the in-situ nursery (in yellow).

Percent net growth among sizes and residence times

Pyramid assays revealed that the in-situ and ex-situ outplanting experiments differed greatly in their observed effects on net growth (Fig. 6). Because it was not possible to obtain permits for a fully reciprocal transplantation design and we have only the local outplanting site for each nursery, we cannot determine whether it is the site, the nursery design, or some interaction of both that contributed to this difference. However, neither the in-situ or ex-situ nursery duration showed consistently dramatic improvement for net percent growth after outplanting, and only M. capitata showed a positive effect of both total time in the ex-situ nursery, and its interaction with size class on overall net growth (Fig. 6 and Appendix 12A). Despite some trends in the data, no such effects were significant for P. compressa (Appendix 12B). For M. capitata at the ex-situ nursery, small fragments experienced the largest net-growth benefit from increased nursery time, but this effect was reduced for the larger size classes. In contrast, neither M. capitata nor P. compressa showed a significant benefit to net growth from prolonged duration in the in-situ nursery (Appendix 13).

3D Structure from motion (SfM)

For both species, there was a strong positive relationship between surface area derived from both the 2D top-down area and 3D modeled approaches (Fig. 3F). Both approaches to measuring the coral colonies resulted in highly similar trends and yielded comparable results according to the generalized linear model (GLM) (Appendices 12 and 13). See Appendix 14 for a further comparison between the two approaches and discussion of the relative time savings from using 3D methods at scale for future efforts. Because the results and interpretation remained unchanged between the 2D and 3D models, and only the pyramid assays showed substantial vertical growth (Fig. 3), we opted for consistency and only present the 2D measurements for both assay types here.

Assay performance

The pyramid assay performed well in both low- and high-flow environments with 94% assay recovery at the in-situ nursery site (low flow) and 74% recovery at the ex-situ outplanting site (high flow). Pyramids were easy to handle and small enough to be mass produced and housed in either nursery. The design also made outplanting extremely easy because their weight allowed for placement in sandy rocky areas at the low-flow in-situ outplanting site and attachment to the reef with epoxy at the high-flow ex-situ outplanting site. Although we did not quantify sediment accumulation directly, visual inspection confirmed that the sloped sides of the pyramids reduced issues with sedimentation while still allowing for a clearly visible label on top during outplanting (Fig. 2). The design provided a weighty solid substrate for fusion of coral tissue (Figs. 3A–3D) without appearing artificial, rapidly blending into the reef substrate and making some difficult to detect by the end of the experiment (Appendix 1C, 1D). However, the small size of each face limited how long the assay could be used to assess individual survivorship, because fragments began to fuse making it hard to distinguish individuals. The compact size also limited the number of colony fragments which could be affixed, which is why the block assay was used to assess spatial variation among individuals.

Block assays: outplanting sites

Fragment survival likelihood among sites

Overall survivorship was 55% for M. capitata and 56% for P. compressa across all sites (Appendix 15). There was significant variation among the species across sites for both survivorship and growth. M. capitata had the lowest survivorship (28%) at site six and the highest survivorship (72%) at sites two and 10. Porites compressa had only 42% survivorship at sites five and eight to a maximum of 72% at site seven (Fig. 4B). There was approximately twice as much variation in survivorship across individual colonies of M. capitata compared to P. compressa (36.4% vs. 12.4%, respectively). For both species a larger proportion of survivorship variance was explained by the fixed effect of ‘site’ while a smaller proportion of variance was attributed to the random effect of ‘genotype’ and there appears to be an interaction because colonies that did among the best and worst at one site would reverse that trend at another. The models of M. capitata and P. compressa explained approximately 32.65% (25.2% marginal), and 11.89% (8.59% marginal) of variation in fragment survivorship, respectively (Appendix 16).

Percent net growth among sites

The growth of both species was significantly different among sites, but M. capitata had more than an order of magnitude higher variation in growth (641% vs. 49.5%) among individuals compared to P. compressa (Appendix 17). Percent net growth increased for both species by over 100% during the 8 months, with a mean 104% increase for M. capitata and 129% for P. compressa (Appendix 15). Even with individual variation, there were consistent differences among sites. The lowest growth rate for M. capitata was at site six (27%), and the highest at site two (155%), whereas for P. compressa the lowest growth was seen at site two (78%) and the highest at site 10 (272%). Looking across the environmental gradient of the bay, with the exception of site five, M. capitata tended to show the lowest growth rates (27–81%) at the central sites 3–8, and highest growth rates (123–155%) at the northern and southern ends of the bay (1–2 and 9–10). P. compressa, by comparison, had no obviously reduced growth regions across the bay, and only the northern sites (9–10) stand out (211–272%) with site 10 P. compressa showing the highest net growth than any other location in the bay (Fig. 4C and Appendix 15).

Assay performance

The block assays were easily deployed without epoxy due to the low wave energy environment in Kānéohe Bay, and all were successfully retrieved without loss in the field. Three top labels were lost, but in each case the backup labels remained attached. Although the flat horizontal surface appeared to retain more sediment than the sloped faces of the pyramid assay, sedimentation was minimal at most sites, and survivorship and growth were of the same magnitude between each design. However, the block design accommodated a larger number of unique colony fragments (n = 18) than the pyramid assays (Fig. 2), while still maintaining sufficient spacing for several months of growth without direct fragment-to-fragment interaction. This design allows for rapid performance testing of individual genotypes in potential restoration locations and could help to identify and focus efforts on which species and individuals are most likely to thrive at a given location.

Discussion

Interest in active coral reef restoration and strategies to scale such efforts has increased dramatically as coral reefs continue to decline globally (Anthony et al., 2017; Hein et al., 2020; Hesley et al., 2017; Omori, 2019; Rinkevich, 2008; Shaver et al., 2020; van Oppen et al., 2015; Vardi et al., 2021; Voolstra et al., 2021). However, most efforts to date remain short-term, small scale, and often lack clear and achievable objectives or rigorous monitoring and reporting about whether those objectives were reached (Boström-Einarsson et al., 2020). Several have pointed out that ecological interactions are rarely considered but critical factors which can affect outplanting success (Boström-Einarsson et al., 2020; Hein et al., 2017; Ladd & Shantz, 2020). Further, Hein et al. (2017) found that 88% of studies published to date use growth and survival of coral fragments as the primary indicators of restoration success, and argue that a more realistic range of ecological indicators along with sociocultural, economic, and governance should all be considered when evaluating the success of reef restoration projects. One such factor is cost-effectiveness of the restoration activities and here we propose short-term assays that can help optimize restoration activities by providing information to maximize survivorship and growth while minimizing nursery and labor costs.

Active restoration of a reef through ‘coral gardening’ or ‘farming’ is generally considered a two-step process: first, raising coral fragments in a nursery to a size that minimizes mortality risk, before second, harvesting and outplanting them to the desired site (Rinkevich, 2006). How long coral fragments need to be raised in the nursery will depend on a variety of factors such as species and growth rate, but fast-growing branching species like Acroporids are usually large enough to outplant within 1 year (Horoszowski-Fridman et al., 2015; Mbije, Spanier & Rinkevich, 2010). Micro-fragmentation, currently accounting for less than 5% of coral transplantation studies to date (Boström-Einarsson et al., 2020), focuses almost entirely on slower growing massive and encrusting species and predominantly using very small fragments (~1 cm2). These small fragments are either grown separately on a plug (part of the reskinning method; C. Page, 2015, Unpublished data) or attached to a module (such as a concrete block) and housed in a nursery until they fuse together prior to outplanting (Forsman et al., 2015; Page, Muller & Vaughan, 2018). Slower growing massive and encrusting species take longer to reach equivalent sizes, so periods of up to 2 years in the nursery have been recommended (dela Cruz et al., 2015; Page, Muller & Vaughan, 2018). Because labor is generally the highest cost of restoration, doubling the nursery time can dramatically increase the cost of such efforts, and likely explains why it is rare to farm slower growing corals (Boström-Einarsson et al., 2020). Here we found that both plating and massive corals responded well to micro-fragmentation and outplanting, and that the net percent gains for both species we selected could be substantially increased over natural growth rates. At the ex-situ nursery we saw the expected relationship between nursery residence time and increased survivorship among the smallest fragments (≈1 cm2) of M. capitata which also had the highest net growth rate overall (Fig. 6). This growth rate was offset somewhat by higher mortality (58%) than medium (33%) or large (30%) fragments (Appendix 9), and culture in the ex-situ nursery had only a weak positive effect on net growth of medium fragments (≈3 cm2) of M. capitata but no benefit at the in-situ nursery or for larger sizes (Appendices 12 and 13). Contrary to expectations, we found no evidence for size-specific benefits for either mortality or growth with nursery duration for P. compressa in either the in-situ or ex-situ nursery. Growing the fragments in a nursery requires space and labor that add to the total cost and reduce scalability of restoration efforts. Consequently, identifying the optimal fragment size that ensures high survivorship will be critical in decreasing the need for coral source material, and reducing labor through minimal handling and nursery residence time is important to optimize cost-efficiency of restoration activities.

The primary benefit of using very small fragments is higher yield with a reduced environmental impact along with increased size-specific growth rates in comparison to larger fragments (Forsman, Rinkevich & Hunter, 2006; Page, Muller & Vaughan, 2018). However, predation can reduce survivorship of these smaller fragments. For example, Koval et al. (2020) found that for four species of massive corals in Florida (Orbicella faveolata, Montastraea cavernosa, Pseudodiploria clivosa, and P. strigosa), fragments <5 cm2 experienced severe tissue damage or complete removal of fragments in the first week of deployment due to corallivorous fish. In Hawai'i, Jayewardene, Donahue & Birkeland (2009) found coral fragments <2 cm2 were entirely removed by corallivorous fish, but nubbins of 4 cm2 or greater were only partially consumed. Likewise, Forsman, Rinkevich & Hunter (2006) found fragments 3 cm2 or larger had the highest survival and growth rates for P. compressa with no evidence of size specific mortality beyond that. Here, we did not observe obvious signs of fish predation (in the form of bite marks), but this will obviously vary among locations depending on the density and species of corallivores. Unfortunately, it was not possible to directly compare the in-situ and ex-situ nurseries, because reciprocal transplantation over these large distances carries risks of vectoring disease or invasive species and are not permitted in the State of Hawai'i. The in-situ nursery site had similar conditions (low water flow, medium sedimentation, ~2 m depth) to the in-situ outplanting site itself. In comparison, the ex-situ nursery site had vastly dissimilar conditions to idealized indoor nursery tank conditions, with high water flow, medium sedimentation, and ~5 m depth. The differences between the nursery and outplanting sites may account for the variation in significant size-specific growth and survivorship among only the smallest M. capitata fragments at the ex-situ site. The value of field assays such as those described herein is to learn such site-specific information in advance of major restoration activities to optimize the efforts.

Pyramid assays provided a cost-effective way to test size-specific survivorship and growth under field conditions. Recovery rates were high in both low and high flow environments, and their small size took up minimal space in the nurseries and made them relatively easy to handle during outplanting. The sloped surfaces minimized sediment accumulation on the coral fragments and the label was easily readable and could be used to orientate the assays during outplanting. Their three-dimensional design provided a suitable substrate for isogenic fusion of fragments on all exposed surfaces, which does not appear artificial and rapidly blends into the reef substrate allowing modules to be left in the field if desired. This rapid overgrowth limits the duration of research use however, because when colonies fuse and grow over the label it makes the modules hard to locate and area cover estimates become much more difficult. The small size also restricts the number of genotypes which could be attached on any given assay. These shortcomings can be offset by the block assays which were purchased from a hardware store, making them a more accessible and cheaper option for some restoration projects. Using blocks allowed us to test 18 genotypes, as opposed to a maximum of three on a pyramid, to assay site-specific differences in individual performance. Additionally, the weight of these blocks meant that they did not need to be attached to the reef even in moderately wave exposed reef areas, and so could be easily placed out and collected for a short-term site assessment.

While there were general trends among sites, with higher overall growth rates in the far southern (1–2) and far northern (9–10) portions of the bay (Fig. 4), we found considerable variation among individual performance at different locations throughout the bay (Appendix 17). Slowly acclimating corals to conditions they will experience in the field could minimize stresses and reduce predation and mortality among outplanted corals (Horoszowski-Fridman et al., 2015; Page, Muller & Vaughan, 2018). However, some traits tend to be less plastic, and corals may never acclimate to a degree that alters survivorship during transplantation (e.g., Barott et al., 2021). Acclimation through similarity to the nursery conditions could explain increased growth at sites one and two which are most similar in terms of being low energy lagoonal habitats with similar hydrodynamic regime, temperature, pH, sedimentation, and nutrient levels, but sites nine and 10 are the most dissimilar to nursery conditions across the environmental gradient of the bay (Bahr, Jokiel & Toonen, 2015; Caruso et al., 2021; Lowe et al., 2009a, 2009b). Sites in the central portion of the bay showed lowest survivorship of M. capitata, whereas P. compressa had similar rates of survivorship across all outplanting locations (Fig. 4B). Because site-specific effects dominate and no single coral is resilient to every pressure faced in every location when outplanting, these results highlight the importance of matching effort to that spatial variation during outplanting. Therefore, rather than trying to acclimate corals to novel site conditions, assays such as these provide an alternative approach to optimize efforts to minimize mortality during restoration activities. Short-term block assays inform which species and individual genotypes have higher survivorship and growth at particular outplanting sites. Pyramid assays on the most successful individuals then allow restoration practitioners to optimize the size and spacing of fragments to maximize survivorship and growth among outplanted corals. By employing a combination of these assays over a period of 2–4 months each, restoration projects could dramatically reduce costs and improve success rates.

Conclusions

It is impossible to generalize methods for all species of corals at all sites, therefore rapid assays such as these are an important step to establish interspecies variation in the performance of variously sized fragments, as well as the role of nursery residence time, and individual performance at a given restoration site. Our study was designed to streamline outplanting practices for two encrusting species in Hawai'i, but more generally simple assays such as these can be used on almost any reef restoration site to identify which species of coral and which individuals are most likely to survive and grow at the outplanting site and on which to focus restoration efforts. The rapid assays we outline here are a simple and highly flexible tool to gather critical preliminary information essential to scaling large restoration projects efficiently and to maximize both the likelihood of success and cost-efficiency. Care however needs to be taken with the assays and the corals attached that they not pose unacceptable environmental risks to the outplanting habitat through overuse and unwanted introduction of invasive species.

Overall, we see relatively little positive benefit of prolonged residence time in either the in-situ or ex-situ nursery. Only the smallest fragments (~1 cm2) of M. capitata showed a significant benefit of nursery residence time on survivorship and growth in the field following outplanting, suggesting that construction and staffing of nurseries may not pay dividends on that investment for large-scale restoration projects. However, this is also the same treatment that showed the greatest overall percent net growth gain in the experiments, highlighting the need for clear objectives for restoration activities. Thus, for a project in which the objective is to outplant in the most cost-effective manner with the greatest coral survival (with the aim that fragments will fuse together in the future), we would recommend considering direct outplanting of medium-sized fragments (~3 cm2) without any nursery care. In contrast, if the objective were to maximize growth for a high value species with limited starting material (such as rare or ESA-listed corals), or outplanting of larger colonies formed from fully fused micro-fragments, then targeting ~1 cm2 fragments raised in an ex-situ nursery with species specific tank conditions would best achieve this goal. Because the goals of each project are likely to differ as much as the species and local environmental conditions that will affect the achievement of those goals, we recommend assays such as these be undertaken to inform efforts to reduce costs and increase productivity prior to undertaking large-scale restoration activity.

Supplemental Information

Supplemental Information 1 Images of pyramid assay donor colonies, and an example of a Hawai'i Coral Reef Nursery (HCRN) larger pyramid assay soon after outplanting and then over time blending in with the surrounding reef.

Images of (A) one of the three Montipora capitata parent colonies from the in-situ nursery (located at the Hawai'i Institute of Marine Biology) and (B) quarantining Porites compressa genotypes 1, 2, and 3 in the ex-situ nursery tanks at Ānuenue Fisheries Research Center (AFRC) Hawai'i Coral Restoration Nursery (HCRN) on Oʻahu, Hawai'i (C) example of a HCRN fully-grown 42 cm pyramid structure with Porites evermanni 1 month after outplanting and (D) the same colony two and half years later (not used in this experiment).

Click here for additional data file.

Supplemental Information 2 Pyramid assay deployment timeline.

Timeline of coral assay deployment. Phase 1 (0 months) set 1 was outplanted on the reef and set 2 and 3 were placed into the nursery (total = 54 assays per nursery). Phase 2 and 3 provided 4 months (for set 2) and 8 months (for set 3) of growth respectively in the nursery prior to transplantation on the reef. Phase 4 was the final growth measurement representing 12 months of growth since fragmentation at the in-situ nursery outplanting site and 17 months for the ex-situ site.

Click here for additional data file.

Supplemental Information 3 Pyramid assay net growth data.

Pyramid assay net growth data by pyramid face for small (7 × 1 cm2 = 7 cm2) medium (3 × 3 cm2 =9 cm2) and large (1 × 9 cm2=9 cm2) fragments of Montipora capitata, and Porites compressa, outplanted from the in-situ and ex-situ nurseries with variable outplanting times (directly after fragmentation, after 4 months and after 8 months).

Click here for additional data file.

Supplemental Information 4 Pyramid assay survivorship data.

Pyramid assay survivorship (alive (1) or dead (0)) datasheet for small (1 cm2) medium (3 cm2) and large (9 cm2) individual fragments of Montipora capitata, and Porites compressa, outplanted from the in-situ and ex-situ nurseries with variable outplanting times (directly after fragmentation, after 4 months and after 8 months).

Click here for additional data file.

Supplemental Information 5 Pyramid assay statistical analyses.

Pyramid assay statistical analyses with Generalized Linear Mixed Models (glmm) examining net growth, and Linear Mixed effects Models (lmm) assessing fragment survivorship based on total time in the nursery and coral fragment size class (fixed effects), while accounting for variation due to parent colony (genotype) and deployment pyramid within colony (random effects) for Montipora capitata, and Porites compressa fragments at the in-situ and ex-situ nurseries.

Click here for additional data file.

Supplemental Information 6 Block assay donor colonies.

The block assay (A) nine Porites compressa, and (B) nine Montipora capitata parent/donor colonies located in the in-situ nursery prior to fragmentation.

Click here for additional data file.

Supplemental Information 7 Block assay survivorship and growth dataset.

Block assay survivorship (alive (1) or dead (0)) and net growth (cm2) datasheet for individual fragments of Montipora capitata, and Porites compressa, outplanted to ten patch reefs in Kānéohe Bay, Hawai'i from the in-situ nursery.

Click here for additional data file.

Supplemental Information 8 Block assay statistical analyses.

Block assay statistical analyses with Generalized Linear Mixed Models (glmm) examining net growth, and Linear Mixed effects Models (lmm) assessing fragment survivorship of Montipora capitata, and Porites compressa fragments across ten patch reef sites as the fixed effect and genotype as a random effect in Kānéohe Bay, Oʻahu, Hawai'i.

Click here for additional data file.

Supplemental Information 9 Pyramid assay survivorship and net growth summary.

Pyramid assay summary table with overall percent survivorship (based on the number of fragments) and percent net growth (based on tissue area (cm2)) for Montipora capitata, and Porites compressa for small, medium, and large fragments at both the in-situ and ex-situ nursery sites. Two-dimensional percent net growth was calculated from top-down planar images taken at both nurseries of those fragments which survived till the end, while three-dimensional growth at the in-situ nursery was based on Structure from motion (SfM) models.

Click here for additional data file.

Supplemental Information 10 Ex-situ nursery pyramid assay model output assessing the likelihood of individual fragments surviving to the end of the experiment based on their total time in the nursery and size class (small, medium, or large).

Summary of generalized linear mixed-effects models (glmm) assessing the likelihood of individual fragments surviving to the end of the experiment based on their total time in the ex-situ nursery and their size class (fixed effects), while accounting for variation due to parent colony (genotype) and deployment pyramid within colony (random effects) for (A) Montipora capitata, and (B) Porites compressa fragments at the ex-situ nursery. Overall model performance was evaluated as adjusted marginal versus conditional R2.

Click here for additional data file.

Supplemental Information 11 In-situ nursery pyramid assay model output assessing the likelihood of individual fragments surviving to the end of the experiment based on their total time in the nursery and size class (small, medium, or large).

Summary of generalized linear mixed-effects models (glmm) assessing the likelihood of individual fragments surviving to the end of the experiment based on their total time in the in-situ nursery and their size class (fixed effects), while accounting for variation due to parent colony (genotype) and deployment pyramid within colony (random effects) for (A) Montipora capitata, and (B) Porites compressa fragments at the in-situ nursery. Overall model performance was evaluated as adjusted marginal versus conditional R2.

Click here for additional data file.

Supplemental Information 12 Ex-situ nursery pyramid assay model output assessing the percentage change in net growth based on their total time in the nursery and size class (small, medium, or large).

Results of pyramid assay linear mixed effects model (lmm) and type 3 ANOVA output assessing percentage change of net growth (on each pyramid face) relative to fragment size (small (1 cm2), medium (3 cm2) or large (9 cm2)) and nursery residence time (fixed effects), while accounting for variation due to parent colony (genotype) and deployment pyramid within colony (random effects) for (A) Montipora capitata, and (B) Porites compressa fragments at the ex-situ nursery.

Click here for additional data file.

Supplemental Information 13 In-situ nursery pyramid assay model output assessing the percentage change in net growth based on their total time in the nursery and size class (small, medium, or large).

Results of pyramid assay linear mixed effects model (lmm) and type 3 ANOVA output assessing percentage change of net growth (on each pyramid face) relative to fragment size (small (1 cm2), medium (3 cm2) or large (9 cm2)) and nursery residence time (fixed effects), while accounting for variation due to parent colony (genotype) and deployment pyramid within colony (random effects) for (A) Montipora capitata, and (B) Porites compressa fragments at the in-situ nursery.

Click here for additional data file.

Supplemental Information 14 Comparison of 2D and 3D measurements of growth.

Text further explaining the difference found between 2D top-down planar image and 3D Structure from Motion (SfM) measurements of coral fragments recorded at the in-situ nursery due to greater arborescent growth here.

Click here for additional data file.

Supplemental Information 15 Block assay survivorship and net growth summary table.

Block assay survivorship and net growth summary table including the number of Montipora capitata and Porites compressa fragments with any live tissue which survived across the 10 outplanting sites in Kānéohe Bay, Oʻahu, from the start to the end of the experiment, along with percentage survivorship and overall live tissue area cover (cm2) from the start and end with overall percent net growth of those fragments which survived till the end.

Click here for additional data file.

Supplemental Information 16 Block assay model output assessing the survivorship likelihood of Montipora capitata and Porites compressa fragments across ten patch reef sites in Kānéohe Bay, Oʻahu, Hawai'i.

Summary of generalized linear mixed-effects models (GLMM) examining survivorship across 10 patch reef sites as the fixed effect and genotype as a random effect in Kānéohe Bay, Oʻahu, of (A) Montipora capitata, and (B) Porites compressa fragments. The estimates represent the likelihood a coral fragment survived from the time of outplanting to the end of the experiment. Overall model performance was evaluated as adjusted marginal versus conditional R2.

Click here for additional data file.

Supplemental Information 17 Block assay model output assessing the net growth of Montipora capitata and Porites compressa fragments across ten patch reef sites in Kānéohe Bay, Oʻahu, Hawai'i.

Results of block assay linear mixed effects model (lmm) and type 3 ANOVA output assessing percentage net growth across 10 patch reef sites in Kānéohe Bay, Oʻahu, of (A) Montipora capitata, and (B) Porites compressa fragments, with genotype as a random effect.

Click here for additional data file.

We’d like to thank all the coral nursery volunteers and students who have been an integral part of the past, present and future productivity of both the HIMB and HCRN coral nurseries. In addition, we couldn’t have completed this work without the fieldwork help of Mykle Hoban, Jan Vicente, Eileen Nalley, and Mariana Rocha de Souza along with the support of the rest of the ToBo lab.

Additional Information and Declarations

Competing Interests

Author Contributions

Field Study Permissions

Data Availability

Robert J. Toonen is an Academic Editor for PeerJ and specifically the special issue “The State of Active Coral Reef Conservation and Restoration”.

Ingrid S. S. Knapp conceived and designed the experiments, performed the experiments, analyzed the data, prepared figures and/or tables, authored or reviewed drafts of the article, and approved the final draft.

Zac H. Forsman conceived and designed the experiments, performed the experiments, authored or reviewed drafts of the article, and approved the final draft.

Austin Greene performed the experiments, analyzed the data, prepared figures and/or tables, authored or reviewed drafts of the article, and approved the final draft.

Erika C. Johnston performed the experiments, analyzed the data, authored or reviewed drafts of the article, and approved the final draft.

Claire E. Bardin performed the experiments, analyzed the data, authored or reviewed drafts of the article, and approved the final draft.

Norton Chan conceived and designed the experiments, authored or reviewed drafts of the article, provided space and expertise in aquariums and pyramid assay construction, and approved the final draft.

Chelsea Wolke conceived and designed the experiments, authored or reviewed drafts of the article, provided space and expertise in aquariums and pyramid assay construction, and approved the final draft.

David Gulko conceived and designed the experiments, authored or reviewed drafts of the article, provided space and expertise in aquariums and pyramid assay construction, and approved the final draft.

Robert J. Toonen conceived and designed the experiments, authored or reviewed drafts of the article, and approved the final draft.

The following information was supplied relating to field study approvals (i.e., approving body and any reference numbers):

Collection and monitoring Special Activities Permits (SAP) were approved by the Division of Aquatic Resources (DAR) at Hawai'i’s Department of Land and Natural Resources (DLNR).

The following information was supplied regarding data availability:

The raw data and R markdown files for the pyramid assays and the block assays are available in the Supplemental Files.

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
