# Peer review of "Coral micro-fragmentation assays for optimizing active reef restoration efforts"

_PeerJ, doi:10.7717/peerj.13653_

## Round 0.1 · original submission · Minor Revisions

My apologies for the slow turnaround but for some reason it took longer than usual to locate responsive reviewers. Your interesting article has now been reviewed by two experts who both ranked the suggested adjustments as “minor”, with which I agree. Reviewer 1 (Dave Vaughan) pointed to a perceived conflict between the statements in lines 423-425 and 480-484 and suggest reworking a statement in lines 650-655. He also suggests updating the literature beyond 2018 (please do!). Reviewer 2 would like to see data on water quality and flow characteristics. Please address any substantive comment by the reviewers. I look forward to receiving the revision.

·

Basic reporting

This article is very good and this type of information is needed in the coral restoration realm. Reporting on the differences in survival and growth for two types of nurseries and for two species is commendable and will be of great interest to the rest of the restoration world. Literature could be more up to date in this industry as items are changing rabidly. There is great backround literature up thru 2018, but there is more relevant literature 2020-2022 available on this subject, including ones from some of the authors.

Experimental design

Experimental design is good, however 3 species and 3 types of substrates would have been better and a longer experimental period for monitoring would have been nice. All in all a good study for getting some results to present

Validity of the findings

All finding are valid, with the exception of one..
In
423 contrast to the ex-situ outplanting site, there were no significant effects of nursery time, size
424 class, or the interaction of these effects for fragment survivorship at the in-situ outplanting site
425 for either coral species

How does that statement compare to this: There was significant variation among the species across sites for both
480 survivorship and growth. M. capitata had the lowest survivorship (28%) at site 6 and the highest
481 survivorship (72%) at sites 2 and 10. P. compressa had only 42% survivorship at sites 5 and 8 to
482 a maximum of 72% at site 7 (Figure 4). There was approximately twice as much variation in
483 survivorship across individual colonies in M. capitata compared to P. compressa (36.4% vs
484 12.4%, respectively).

Additional comments

I would re work this statement.
Thus, for a project in which the objective is to outplant in the
650 least expensive and most cost-effective manner with the greatest coral survival outcome, for
651 these species at these sites we would recommend direct outplanting of medium-sized fragments
652 (~3cm2
) without any nursery care. In contrast, if the objective were to maximize growth for a
653 high value species (such as rare or ESA-listed corals) in which starting material was limited or
654 outplanting of large sexually reproducing colonies was the objective, then targeting ~1cm2
655 fragments raised in the ex-situ nursery would best achieve this goal

Line 650 makes it sound like the line 651 is the recommendation for least expensive, cost effective, survival, etc.
Line 653-655 makes i sound like this goal is only reached for the exception of lines 653/4

the option of all nurseries outplanting the optimal size (medium, etc) should all look at starting with 1cm2 or less as a suggestion, not as a limited exception.

Also recommend these recent publications;
Land and Field Nurseries: D.Vaughan, 2021, In Active Coral Restoration, JJRoss Pub
Coral Fusion: Harnessing Coral Clonality for Reef Restoration: Forsman, Page and Vaughan 2021 in
Active Coral Restoration, JJRoss pub
Growth and Cyclin-E Expression in the Stony Coral Species Orbicella faveolata Post- Microfragmentation, Deanna Sopher, et.al 2022 Biological Bulletin
Vaughan, D., Tiecher, S., Halpern. G. and Oliver, J. 2019. Building more resilient coral reefs through new marine technologies, Science, and Models. Marine Technology Society Journal Vol 53:5

Reviewer 2 ·

Basic reporting

Figure 6.

1. Using black dots and lines on a dark background makes this figure nearly impossible to see. Either lighten the background color or change the black dots and lines to white or yellow.

2. Since 62.5% of the population that has color blindness has a form of red-green colorblindness, I would suggest replacing the use of green blocks in the bottom graph of figure 6 with another color such as yellow.

Experimental design

no comment

Validity of the findings

no comment

Additional comments

1. I would like to see data on water quality and flow characteristics of the two out planting sites so that a data driven comparison of the two locales could be made. There are references given that mention that that there are differences but there is not data presented to show how the two sites differ. The difference in growth form for instance is related to water flow and light penetration since the HIMB site is more turbid and lagoon like compare to the more wave exposed site off of Sand Island but there is no data provided to backup this assertion.

---

## Round 0.2 · accepted · Accept

Thank you for addressing the reviewers’ comments. I look forward to seeing your very interesting paper published.